# Choosing an Optimal Sample Preparation in *Caulobacter crescentus* for Untargeted Metabolomics Approaches

**DOI:** 10.3390/metabo9100193

**Published:** 2019-09-20

**Authors:** Julian Pezzatti, Matthieu Bergé, Julien Boccard, Santiago Codesido, Yoric Gagnebin, Patrick H. Viollier, Víctor González-Ruiz, Serge Rudaz

**Affiliations:** 1Institute of Pharmaceutical Sciences of Western Switzerland (ISPSO), University Medical Centre, 1206 Geneva, Switzerland; julian.pezzatti@unige.ch (J.P.); julien.boccard@unige.ch (J.B.); santiago.codesido@unige.ch (S.C.); yoric.gagnebin@unige.ch (Y.G.); victor.gonzalez@unige.ch (V.G.-R.); 2Department of Microbiology and Molecular Medicine, Faculty of Medicine, University of Geneva, University Medical Centre, 1206 Geneva, Switzerland; matthieu.berge@unige.ch (M.B.);; 3Swiss Centre for Applied Human Toxicology, 4055 Basel, Switzerland

**Keywords:** metabolomics, sample preparation, hydrophilic interaction liquid chromatography, ion mobility spectrometry, high resolution mass spectrometry, design of experiments, AMOPLS

## Abstract

Untargeted metabolomics aims to provide a global picture of the metabolites present in the system under study. To this end, making a careful choice of sample preparation is mandatory to obtain reliable and reproducible biological information. In this study, eight different sample preparation techniques were evaluated using *Caulobacter crescentus* as a model for Gram-negative bacteria. Two cell retrieval systems, two quenching and extraction solvents, and two cell disruption procedures were combined in a full factorial experimental design. To fully exploit the multivariate structure of the generated data, the ANOVA multiblock orthogonal partial least squares (AMOPLS) algorithm was employed to decompose the contribution of each factor studied and their potential interactions for a set of annotated metabolites. All main effects of the factors studied were found to have a significant contribution on the total observed variability. Cell retrieval, quenching and extraction solvent, and cell disrupting mechanism accounted respectively for 27.6%, 8.4%, and 7.0% of the total variability. The reproducibility and metabolome coverage of the sample preparation procedures were then compared and evaluated in terms of relative standard deviation (RSD) on the area for the detected metabolites. The protocol showing the best performance in terms of recovery, versatility, and variability was centrifugation for cell retrieval, using MeOH:H_2_O (8:2) as quenching and extraction solvent, and freeze-thaw cycles as the cell disrupting mechanism.

## 1. Introduction

Untargeted metabolomic approaches aim to analyze the metabolic composition of biological samples in a holistic manner, with the goal of retrieving and storing as much chemical information as possible. To this end, adequate sample preparation is one of the most critical steps to obtain reliable biological insights [1]. Moreover, for an optimal interpretation of the information gathered from such systems, the annotation of metabolites of interest is a requirement. Indeed, the recent advances in liquid chromatography (LC) coupled with high resolution mass spectrometry (LC-HRMS) and the development of in-house and external databases have made possible the identification and relative quantification of an increasing number of metabolites in many types of biological samples [2,3].

When analyzing polar metabolites, techniques with separation mechanisms that are orthogonal to those of the classical reversed phase LC (RPLC) are needed [4]. Among these, hydrophilic liquid interaction chromatography (HILIC) has proven to be an attractive strategy for such classes of compounds and is well referenced in the literature [5,6,7,8,9,10,11]. Several groups have recently demonstrated that different HILIC stationary phases enable the analysis of many chemical classes of metabolites [12,13,14]. It has been shown that polymeric columns with zwitterionic stationary phases are interesting for the analysis of compounds related to the tricarboxylic acid cycle (TCA) [14].

To perform metabolite annotation in untargeted metabolomics, several criteria must be fulfilled to provide a certain level of confidence to the putatively identified compounds, as defined by the metabolomics standards initiative (MSI) [15]. To maximize the reliability of the identification of features, the retention time, exact mass, isotopic pattern, and MS/MS pattern must be used to match compound properties to those of standards measured under identical experimental conditions [16,17]. With the advent of the latest generation of quadrupole time-of-flight instruments (QTOF), coupled with ion mobility (IM), peak capacity and dynamic range have been increased, enabling to perform all ion fragmentation (AIF) approaches and including the measurement of collision cross section (CCS), which offers another structural information for identification [18,19]. Indeed, CCS values provide valuable physicochemical information about the ions such as three-dimensional conformations [20,21]. Furthermore, the IM technology allows to selectively isolate coeluting peaks resulting in cleaner MS and MS/MS spectra [22].

In untargeted metabolomics experiments, it has already been demonstrated that various sample preparation procedures result in very different metabolite recovery rates, regardless of the biological matrix studied, e.g., human plasma [23,24,25,26,27], mammalian cell cultures [28,29,30,31], and microbial cell cultures [32,33,34,35]. During sample preparation, subtle modifications of the conditions, such as slight changes in the composition of the extraction solvents, can have remarkable effects on the recovery of some metabolites [36]. In vitro metabolomic studies on cell cultures offer many advantages over more complex biological systems including an easily controlled environment, greater reproducibility, lower cost, and easier method transfer to external laboratories for validation [30]. However, when it comes to microbial cell cultures, sample preparation faces challenges such as fast metabolite turnover, degradation, leakage, and poor extraction reproducibility [30,37,38]. These cell cultures require harsher sample processing procedures than body fluids. Metabolite leakage must be prevented during the cell retrieval step [38]. While metabolite losses can be minimized by reducing the number of steps and simplifying the protocol, reproducible sample preparation procedures are necessary to ensure an accurate determination of small changes in metabolite levels [39]. As a rule of thumb, cell metabolism must be quenched quickly, and metabolites extracted rapidly, non-selectively, and reproducibly upon cell sampling [39]. Quenching and extraction steps are of upmost importance in the sample preparation process to obtain metabolic profiles which are representative of the physiological status at the time of sampling [30].

In this study, we aimed to perform a systematic evaluation of the influence of different sample preparation conditions on the nature and variability of the recovered metabolomic profile in Gram-negative bacterium *Caulobacter crescentus*. The latter was chosen as a model for Gram-negative bacteria because its central carbon metabolism, closely linked to cell cycle regulation and cell division, remains to be fully understood [40]. After a comprehensive evaluation of the literature on sample preparation for bacterial cell cultures [32,33,34,35], it was found that no systematic evaluation of different protocols supported by a design of experiment (DOE) approach was available. In the present study, two techniques of cell retrieval, two quenching and extraction solvents, and two cell disrupting mechanisms were investigated. Considering a full factorial design (FFD), eight different sample preparation protocols were tested for the relative quantification of polar metabolites from diverse chemical classes. The simultaneous investigation of several experimental factors constitutes a powerful approach to assess the impact of the metabolites extraction capabilities of each studied combination. Even if principal component analysis (PCA) constitutes often a potent tool to detect relevant variables and patterns [41], an approach dedicated to fully exploit the data structure of complex omics data generated from DOE, i.e., analysis of variance multiblock orthogonal partial least squares (AMOPLS) [42,43,44], was used to assess the contribution of each studied factor on the annotated metabolites.

## 2. Materials and Methods

### 2.1. Chemicals

UPLC-MS grade acetonitrile (ACN), methanol (MeOH), and water (H_2_O) were obtained from Fisher Scientific ( Loughborough, UK). UPLC-MS grade formic acid (FA) was supplied by Biosolve (Valkenswaard, Netherlands). Chloroform (CHCl_3_) was provided by Acros Organics (Geel, Belgium). Ammonium hydroxide and ethylenediaminetetraacetic acid disodium salt dehydrate (EDTA) were obtained from Sigma-Aldrich (Buchs, Switzerland). The major mix IMS/T of calibration kit and leucine-encephalin were acquired from Waters (Milford, MA, USA).

### 2.2. Bacterial Sample Preparation

The eight evaluated sample preparation protocols are summarized in Table 1. Two cell retrieval procedures, two systems of quenching and extraction solvents, and two cell disrupting mechanisms were compared.

#### 2.2.1. Bacterial Cell Cultures and Cell Retrieval

*C. crescentus* cells were grown overnight at 30 °C in a PYE rich medium (2 g/L bactopeptone, 1 g/L yeast extract, 1 mM MgSO_4_, and 0.5 mM CaCl_2_). Two cell retrieval methods were tested, the first one in a liquid medium, and the second as an adaptation of a filter culture-based method [40].

For the liquid culture, overnight cultures were diluted to reach OD_600nm_ ~ 0.4 and 10 mL of cells were centrifuged at 8000 g for 5 min at 4 °C, then cells were resuspended in the desired precooled quenching and extracting solution. For metabolite extraction from filter cultures, overnight cultures were diluted at 30 °C in PYE medium until reaching OD_600nm_ ~ 0.2. Then, 10 mL of culture were transferred onto 0.22 μm mixed cellulose ester (MCE) membrane filter (Millipore) by vacuum filtration. The filters were then deposited on the surface of PYE-agar made with the same growth medium, and the cells were allowed to continue growing at 30 °C for 3 h. Finally, filters were dropped directly into desired precooled quenching and extraction solution.

#### 2.2.2. Quenching and Extraction Methods

The two cold solvents (−20 °C) for quenching and extraction were respectively MeOH:H_2_O 8:2 and MeOH:H_2_O:CHCl_3_ 7:2:1 + EDTA 1 mM. For each bacterial culture sample, 1 mL of solvent was added.

#### 2.2.3. Cell Disruption

Two lysis methods were compared, one by freeze and thaw (F/T) and another by bead beating. Cells were subjected either to lysis by 5 cycles of F/T (−80 °C/40 °C) or to mechanical lysis using 0.1 mm zirconia beads in a Fast prep-24 (MP) disruption instrument (5 cycles of 15” on, 1’ off) maintained at 4 °C. In both cases, cellular debris were removed by centrifugation at 14,000 g, 20 min at 4 °C. Metabolite extracts were kept at −80 °C.

#### 2.2.4. Extract Preparation for LC-MS Analysis

The supernatants were collected, evaporated to dryness using a SpeedVac (ThermoFisher, Langenselbold, Germany) and reconstituted in 100 µL ACN:H_2_O 50:50. Quality control (QC) and diluted QC (dQC) samples were respectively prepared by pooling equivalent volumes of all reconstituted samples and by 1:1 dilution of a certain amount of the QC pool with ACN:H_2_O 50:50. QC and dQC were injected at regular intervals throughout the LC-MS analyses to assess analytical variability.

### 2.3. Liquid Chromatography Conditions

For the LC experiments, a Waters H-Class Acquity UPLC system composed of a quaternary pump, an autosampler including a 15 μL flow-through-needle injector for which temperature was set at 7 °C, and a two-way column manager (Waters, Milford, MA, USA) were used. The injected volume was 10 μL. HILIC separations were performed on a Merck SeQuant ZIC-pHILIC column (150 × 2.1 mm, 5 μm) with the appropriate guard kit (Merck KGaA, Darmstadt, Germany). Solvent A was ACN and solvent B was H_2_O containing 2.8 mM ammonium formate adjusted at pH 9.00. The pH of the solution was checked and found to be stable during one week at room temperature. Column temperature and flow rate were set at 40 °C and 300 µL min^−1^, respectively. The gradient elution was as follows: 5% B for 1 min, increasing to 51% B during 9 min, holding for 3 min at 51% B, and then returning back to 5% B in 0.1 min and re-equilibrating the column for 6.9 min.

### 2.4. Mass Spectrometric Conditions

HRMS was carried out on a Vion TWIMS-QTOF (Waters, Manchester, UK) equipped with an ESI source. Analyses were performed in negative ESI mode to acquire continuum data in the range of 50–1000 *m/z* with a scan time of 0.2 s. The source parameters were set as follows: capillary voltage was −2.0 kV, source and desolvation temperatures were set at 120 and 500 °C, respectively, cone and desolvation gas flow were 50 and 800 L/h, respectively. Velocity and height of StepWave1 and StepWave2 were set to 300 m/s and 5 V and to 200 m/s and 30 V, respectively. The high definition MS^E^ (HDMS^E^, using ion mobility) settings consisted of trap wave velocity at 100 m/s; trap pulse height A at 10 V; trap pulse height B at 5 V; ion mobility spectrometry (IMS) wave velocity at 250 m/s; IMS pulse height at 45 V; wave delay set at 20 pushes, and gate delay at 0 ms. Gas flows of ion mobility instrument were set to 1.60 L/min for trap gas, and 25 mL/min for IMS gas. Buffer gas was nitrogen.

Fragmentation was performed in HDMS^E^ mode. For the collision energy, 6.0 eV was used for low energy and high energy was a ramp from 10 to 60 eV. Nitrogen was used as collision gas. Leucine-encephalin served as a lock mass (554.2615 *m/z* for ESI-) infused at 5 min intervals. The CCS and mass calibration of the instrument were done with the calibration mix “Major mix IMS-TOF calibration” (Waters, Manchester, UK).

### 2.5. Analysis of Raw Data

Chromatogram alignment, peak picking, adduct deconvolution, and feature annotation were sequentially performed on Progenesis QI v2.3 (Nonlinear Dynamics, Waters, Newcastle upon Tyne, UK). The following tolerances were used for feature annotation with regard to a set of pure reference standards (MSMLS Library of Standards, Sigma-Aldrich) measured in the same instrument: 2.5 ppm for precursor and fragment mass, 10% for Rt, and 5% in the case of CCS. Data pretreatment was performed with SUPreMe, an in-house software with capabilities for drift correction, noise filtering, and sample normalization. Finally, data were transferred to SIMCA-P 15.0 software (Umetrics, Umea, Sweden) to perform Principal Component Analysis (PCA). AMOPLS analysis was conducted after unit variance scaling as previously described [42] under the MATLAB^®^ 8 environment (The MathWorks, Natick, MA, USA). A series of 10^4^ random permutations was performed to validate the AMOPLS model and assess the statistical significance of the effects.

## 3. Results and Discussion

Bacterial cell cultures are an interesting resource to evaluate how a given system reacts upon the deletion of certain genes, modifications in the conditions of the growing media, or changes happening in the cell morphogenesis during the cell growth and proliferation processes. To increase our understanding of the endogenous metabolic changes appearing in such biological systems, untargeted metabolomics has already proven to be a valuable approach [39,40,45].

To study the effects of each sample preparation procedure on the extraction of polar metabolites, three parameters, namely cell retrieval, quenching and extraction solvent, and cell disruption, were systematically investigated by two methods. Medium filtering and centrifugation were compared for cell retrieval, cold MeOH:H_2_O 8:2 and cold MeOH:H_2_O:CHCl_3_ 7:3:1 + EDTA 1 mM were compared as two quenching and extraction solvents and, finally, two cell disruption mechanisms (bead beating and freeze-thaw cycles) were tested. In total, the overall performance of eight different procedures was evaluated and compared against the others for the metabolomic analysis of *C. crescentus* cell cultures (Table 1).

The cell retrieval procedures were chosen on the basis of (i) their ability to separate the cells from media, (ii) their ease of application, and (iii) their recognized suitability for the determination of intracellular metabolites [46]. The quenching and extraction solvent mixtures were chosen due to their aptitude to (i) freeze the metabolic status after cell harvesting, (ii) inactivate intracellular enzymes and extract at the same time metabolites, and (iii) extract hydrophilic compounds as widely as possible with the best recoveries [46]. CHCl_3_ was studied because of its potential usefulness to enhance cell wall disruption and to inactivate enzymes, and EDTA due to its ability to chelate metal ions preventing metabolite oxidation and degradation [30,35,37]. Cell disrupting mechanisms were selected due to their capacity to (i) disrupt cell walls and facilitate metabolite extraction [30], (ii) complete the denaturation of all enzymes to avoid further metabolite interconversions, and (iii) prevent significant degradation and chemical conversion of the extracted metabolites [47]. Finally, metabolomic analyses were performed in HILIC mode coupled to a Vion TWIMS-QTOF operating in negative ESI polarity. The choice of this UHPLC-HRMS platform enables the investigation of highly polar metabolites such as those related to the energy metabolism with excellent performance, as described in our previous paper [14]. Of course, the use of other LC-MS platforms to analyze samples issued from the same experimental design would provide different results, since the performance of each method to retain and ionize different classes of metabolites will be different.

### 3.1. Evaluation of the Extraction

The first goal of the investigation was to assess the recovery and the overall variability on annotated metabolites. Analyses were performed in triplicate for each combination of conditions and evaluated with the help of various procedures such as relative standard deviation (RSD) and multivariate analysis (MVA). PCA was used to evaluate the similarity and dissimilarity of the different extraction conditions, while AMOPLS was used to estimate the contribution of experimental factors to variations annotated metabolites abundances.

The performance reached by each sample preparation protocol was first assessed on the overall set of features remaining after data pretreatment. From the raw dataset, more than 8500 features were found in the initial peak-picking step. Data pretreatment included filtering of the features based on QC-to-dQC peak area ratio and RSD in QC samples, LOESS drift correction by interpolation with the QC samples, and probabilistic quotient normalization (PQN) methods [48]. These respectively eliminate noisy features, alleviate the changes in the per-metabolite variations of the instrument’s response factor, and fix any sample normalization issues remaining from the sample preparation. In particular, LOESS has the advantage of being nonparametric, so no external information needs to be introduced, which would unnecessarily increase variability. The only requirement is the presence of QC samples. PQN was chosen for similar reasons, since it is highly robust and does not depend either on external information that would introduce uncertainty into the correction. Its only drawback is the need for enough variables to derive the ratio distribution, but that need is more than fulfilled in untargeted analyses such as the present one. The whole pretreatment procedure reduces the analytical variability in the data, so as to present a dataset to the statistical analyses, where as much variability as possible is of biological origin. Altogether, a total of 904 filtered and corrected features were found to be analytically reliable across all the samples in the experimental design, and they were retained for the posterior analysis (Appendix A).

### 3.2. Metabolites Annotation

For the purpose of completing the LC-MS untargeted workflow of such experiments, feature annotation must be used to assess which metabolites are preferentially retrieved by each sample preparation procedure, thus enabling the study of certain metabolic pathways in future experiments with *C. crescentus*. In the present study, features were first annotated by using their *m/z* and retention time [15] and their identities were then confirmed by MS/MS or CCS data. MS/MS acquisitions were carried out in all ion fragmentation (AIF) mode by using an energy ramp (10–60 eV). The AIF mode presents the potential that a simultaneous acquisition of both precursor and product ions can be considered in the same analytical run. However, AIF suffers from inaccuracies, which can happen when LC coeluting peaks with similar peak shapes possess similar MS/MS spectra (common product ions). This limitation can be overcome by (i) computational deconvolution algorithms that link the precursor ions (at low energy) and the product ions (at high energy) [49] and (ii) the IMS technology, i.e., HDMS^E^, which allows the separation of LC coeluting precursor ions on the drift time dimension before fragmentation. Therefore precursor and product ions can be IMS-aligned, allowing to filter out fragments that do not match the precursor’s drift time [18]. Moreover, IMS offers the potential to gain further structural information by providing CCS values, and therefore a gain in confidence in the metabolites annotation. An example of how both precursor and product ions can be cleared with the help of the IMS technology is presented in Figure 1 with the example of acetyl-CoA. Cleaner MS/MS also reduces the false positives rate during the annotation step. Among the 904 kept features, 133 of them were annotated with the help of *m/z* and retention time and 106 were confirmed by MS/MS or CCS values (Appendix A). This result highlights the fact that false positive annotations (27 unconfirmed annotations) can occur when they are not validated by other orthogonal molecular descriptors. Most of the 106 identified metabolites (almost 55% of the confirmed compounds) belong to one of the following chemical groups: amino acids, nucleotides, and organic acids [50], as presented in Figure 2. This shows the ability of the developed analytical method to detect compounds related to fundamental metabolic pathways in biological samples, such as the energy metabolism.

### 3.3. AMOPLS for Assessing the Contribution of Each Studied Factor to the Overall Variability

To obtain an overview of the major sources of variability in the dataset, unsupervised data analysis was first performed on the 106 annotated metabolites, as measured from the eight sample preparation procedures. According to the PCA score plot presented in Figure 3, the sample preparation approaches that appear well clustered, indicating low intragroup variability, are: centrifugation-CHCl_3_-F/T, centrifugation-MeOH-F/T, and filter-CHCl_3_-F/T. An important difference in the sample preparation between centrifugation and filtering can also be highlighted. Both centrifugation-CHCl_3_-F/T and centrifugation-MeOH-F/T are separated, while filter procedures are regrouped at the top-left region of the PCA.

As a second step of data analysis, AMOPLS was used to get a deeper insight into the specific contribution of each experimental factor (i.e. main effects), and their potential interactions, to the total observed variability. This approach allows us to decompose, quantify, and evaluate the significance of each effect in large datasets issued from full factorial designs. The proposed experimental setup allows the simultaneous evaluation of the three experimental factors under study, i.e., the cell retrieval procedure, the quenching and extraction solvent, and cell disrupting mechanisms, as well as their interactions. The interpretation of the signal variations is performed using effect-specific scores and loadings.

The results of the AMOPLS model are presented in Table 2. All main effects had a significant contribution to the total observed variability, but no interaction between them was found to be statistically significant. The cell retrieval procedure accounted for 27.6% of the total variability, the quenching and extraction solvent for 8.4%, and the cell disrupting mechanism for 7.0%. Remarkably, the cell retrieval procedure (filter/centrifugation) is the most influential parameter. The residuals accounted for 57.0% of the total observed variability, suggesting an important contribution of other sources of variation not formally considered in the experimental design and possibly coming from the biological variability or the sample preparation itself.

The model was then interpreted with the help of the distribution of the observations (groupings) and the contribution of the variables to the specific predictive component associated with each main effect. The score plots are presented in Appendix A, generated with data coming from Appendix A. Squared variable importance in the projection (VIP^2^) values were then calculated (Appendix A) and taken as the quantitative measure of the contribution of all modelled effects coming from a single variable [44]. VIP^2^ values are helpful to rank compounds and spot the ones that play a major role in the model for each studied effect. Figure 4 presents the VIP^2^ values for each studied effect, for the 50 most relevant metabolites out of the 106 annotated ones. Larger VIP^2^ values indicate which effect(s) are the most important ones for each annotated metabolite. In Figure 4, metabolites are ranked according to the importance of the cell retrieval effect and compounds on the left side are those for which this procedure had the strongest impact (largest VIP^2^ values). On the contrary, this effect exerted little influence on metabolites appearing towards the right side of the plot (Appendix A). Some metabolites belonging to the groups of sugars (2-deoxy-glucose, glucose, and xylose) and nucleosides (cytidine, guanine, and xanthine) are among the most influenced ones. Besides the scores and corresponding loading plots (Appendix A, representing the data from Appendix A, respectively), these metabolites were found in higher abundance in the filtering procedure. Secondly, highly polar metabolites related to the amino acid family (L-histidine, L-glutamine, and L-asparagine), phosphorylated sugars (ribose-5-phosphate, glucose-6-phosphate, fructose-1, and 5-bisphosphate) and nucleotides (5’-CMP, IDP, IMP, and GTP) were highly influenced by the quenching and extraction solvent. Probably, the polarity of the used solvents accounts for these observed results, since MeOH:H_2_O 8:2 is more polar than the second system used, thus enhancing the recovery of more polar molecules. Indeed, by looking at the scores and corresponding loading plots (Appendix A), almost all metabolites given as examples were found in higher abundance in MeOH:H_2_O extracting solvent, except IDP, GTP, and fructose-1,5-bisphosphate which were found in higher abundance when CHCl_3_ was present, an observation that can be explained by an enhanced enzyme-denaturing process expected when this solvent was added. Finally, the effect of cells lysis was studied. Metabolites related to lipids (petroselinic acid, palmitic acid, and stearic acid) and nucleotides (ADP, CTP, GDP, CDP, GTP, and ATP) were the most influenced chemical classes. All of these metabolites were found in higher abundance in the F/T procedure except for palmitic acid and petroselinic acid (Appendix A). One hypothesis to explain these results would be the fact that both cell disrupting mechanisms are quite different in terms of the amount of energy applied to the sample. Indeed, beads beating is a harsher procedure compared to F/T cycles and metabolites amenable to degradation via hydrolysis, decarboxylation or oxidation could be more easily impacted.

### 3.4. Assessing the Recovery and the Variability of the Annotated Metabolites

A larger number of metabolites were found in higher abundance with the following combination of factors: cell filtering, MeOH:H_2_O, and F/T cycles (Appendix A). First, nucleosides (guanine, guanosine, cytidine, inosine, xanthosine, and xanthine) were more abundant in sample preparations starting with a filtering step, regardless of what other following procedures were used. One explanation could be that these metabolites leak from cells more easily during the longer centrifugation procedure, as it has already been found for *Escherichia coli* [51]. Second, the MeOH:H_2_O condition enables the retrieval of more metabolites in higher proportion. This can be explained by the medium’s polarity, since this solvent enables a better extraction of polar compounds as compared to the CHCl_3_ system. However, it was expected that the addition of CHCl_3_ and EDTA would prevent metabolite turnover, and oxidation and degradation by metals for nucleotides, CoA, and CoA thioesters derivatives, as described in the literature [37]. This is confirmed for ATP, succinyl-CoA, and CoA which are found in higher proportion in the presence of CHCl_3_ (Appendix A). Interestingly, ADP was found in higher proportion in MeOH. This result supports that in these conditions ATP gets hydrolyzed into ADP, confirming that EDTA and CHCl_3_ prevents the degradation of ATP into ADP. For the cell disruption mechanisms, a large majority of metabolites were found in higher abundance in the F/T cycles procedure (Appendix A). Indeed, some nucleotides, CoA and CoA thioester derivatives, and some organic acids (citric acid), which are highly relevant indicators of the metabolic status of the cells, are found more abundantly when F/T was applied. This could be explained by the facilitation of non-enzymatic degradations in the case of bead beating, as previously discussed. Finally, acetyl-CoA, CoA, TMP, and DAMP, were more abundant in either centrifugation-CHCl_3_-F/T or centrifugation-MeOH:H_2_O-F/T. It can be seen that all four metabolites are more abundant after using centrifugation as cell retrieval, MeOH:H_2_O (except for CoA, found at the same abundance for both solvent systems used) and F/T cycles (except for TMP and DAMP, found at the same abundance for both cells disrupting mechanisms used). These results support that either centrifugation-CHCl_3_-F/T or centrifugation-MeOH:H_2_O-F/T conditions should be chosen in the case of interest towards these particular metabolites.

The recovery obtained using each sample preparation procedure was assessed on the 106 confirmed metabolites and results are shown in Appendix A. Metabolite integration was manually curated among the features of the preprocessed data and considered as undetected when an intensity value of zero was found (total absence). The recoveries of each sample preparation varied from 88% to 95%. It has been noticed that for each different sample preparation, more metabolites were recovered when MeOH:H_2_O solvent was used as compared to MeOH:H_2_O:CHCl_3_ + 1 mM EDTA. The sample preparation Filter-MeOH:H_2_O-F/T had the highest recovery percentage (95%) and centrifugation-CHCl_3_-F/T yielded the lowest recovery percentage (88%). Appendix A also shows that more metabolites can be recovered when the filtering procedure is used, as opposed to centrifugation.

On the other hand, the RSD is crucial since the most repeatable procedures will guarantee data of higher quality in further metabolomics experiments. It was possible to calculate the RSD on the area of each metabolite obtained in triplicate. These results are presented in Figure 5. The RSD values (X) were classified into five categories: 0 < X ≤ 10%, 10 < X ≤ 20%, 20 < X ≤ 30%, 30 < X ≤ 40%, or >40%. The sample preparation extracting the largest number of metabolites, i.e., filter-MeOH:H_2_O-F/T, presents the worst RSD (>40%) results. On the other hand, we found sample preparation such as centrifugation-CHCl_3_-F/T and centrifugation-MeOH:H_2_O-F/T, encompassing RSDs of 0 < X ≤ 10%, 10 < X ≤ 20% and 20 < X ≤ 30% for the highest number of metabolites. Therefore, we would recommend use of the latter two sample preparation procedures when the metabolite extraction rate must be well balanced with repeatability.

## 4. Conclusions

It is currently recognized that sample preparation is a crucial step in untargeted metabolomics workflows as biological outputs will only be drawn from retrieved metabolites. In this work, eight different sample preparation protocols for Gram-negative bacteria *C. crescentus*, combining two cell retrieval systems, two quenching and extraction solvents, and two cell disruption procedures were investigated. The use of a full factorial design to evaluate the contribution of various factors, with the help of an AMOPLS model, provided a potent way to examine their effect on the metabolite recovery rates. Moreover, annotated metabolite variability was assessed with the help of univariate data analysis via the RSD of the area of each metabolite.

The major conclusions about the studied experimental parameters are:(a)For cell harvesting, the centrifugation procedure could lead to a higher number of metabolites leaking out of the cells as compared to filtering, as the latter is faster, preventing metabolite turnover and degradation and leakage which could take place during centrifugation.(b)For quenching and extraction solvent, the combination MeOH:H_2_O enables a better extraction of polar metabolites.(c)For cell disruption, bead beating could lead to higher temperature and more degradation because of its rougher nature. The mechanical disruption obtained by this method is often very advantageous when dealing with tissue fractions, but unnecessary in the studied samples, as it leads only to a larger degradation rate.

To summarize, even if filtering enables the retrieval of more metabolites with higher abundances, RSD evaluation demonstrated a detrimental extraction variability for any of the conditions, as compared to the centrifugation procedures. Therefore, centrifugation should be the selected method for cell retrieval. For the quenching and extraction solvent, the choice should be driven by the needs of the study regarding the specific nature of the metabolites to extract. If the target would be acetyl-CoA, an important metabolite for many biological pathways, MeOH should be used. The F/T cycles retrieve more metabolites at higher abundance and present much better RSD values as compared to bead beating for the cell disrupting mechanism. Finally, the combination of centrifugation-MeOH-F/T appears to perform the best among those evaluated in the present study.

## Figures and Tables

**Figure 1 metabolites-09-00193-f001:**
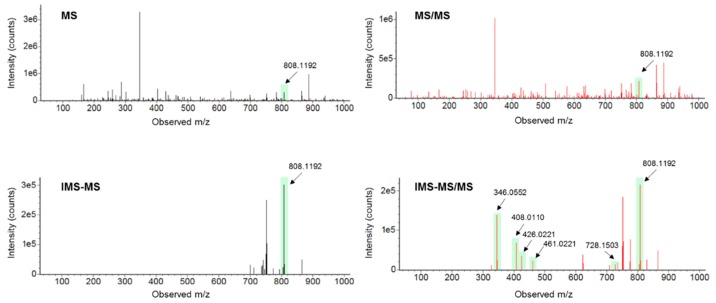
Example of MS and MS/MS spectra without and with IMS filtration for the precursor ion of acetyl-CoA (M-H). Precursor ion and fragment ions are highlighted in filled bars. Cleaner MS and MS/MS spectra are obtained thanks to the IMS separation. Fragment ions were matched to the MS/MS spectra of the chemical standard analyzed under the same analytical conditions.

**Figure 2 metabolites-09-00193-f002:**
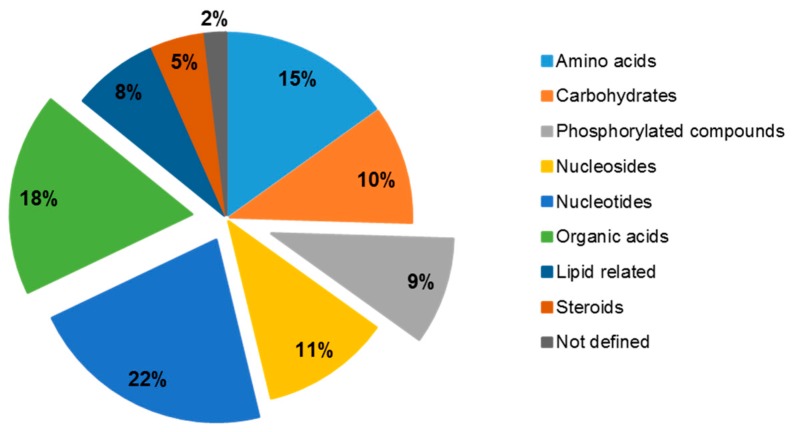
Annotated metabolites chemical groups. Percentage based on the total number of annotated metabolites (106).

**Figure 3 metabolites-09-00193-f003:**
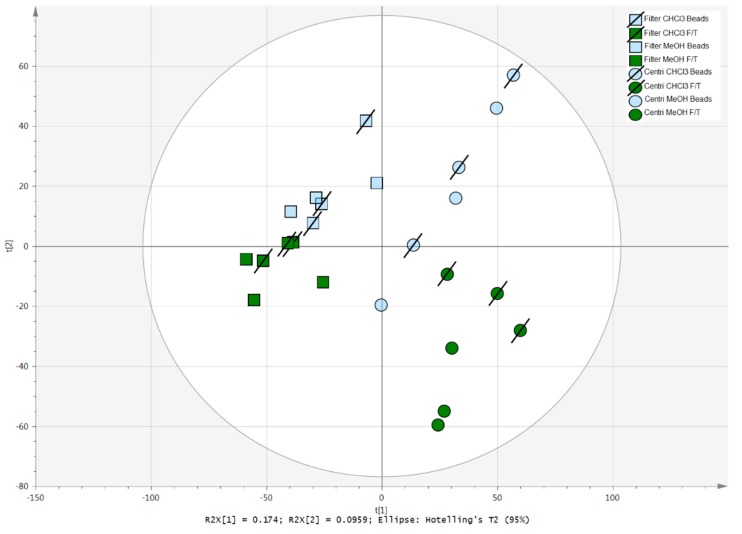
Principal component analysis (PCA) score plot of the eight sample preparation procedures evaluated in triplicates.

**Figure 4 metabolites-09-00193-f004:**
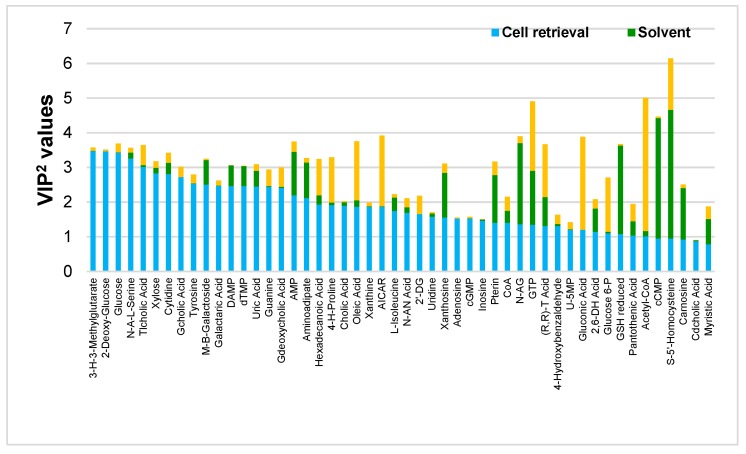
Effect-specific variable importance in the projection (VIP)^2^ values for the 50 out of the 106 annotated metabolites ranked according to the impact of the cell retrieval effect.

**Figure 5 metabolites-09-00193-f005:**
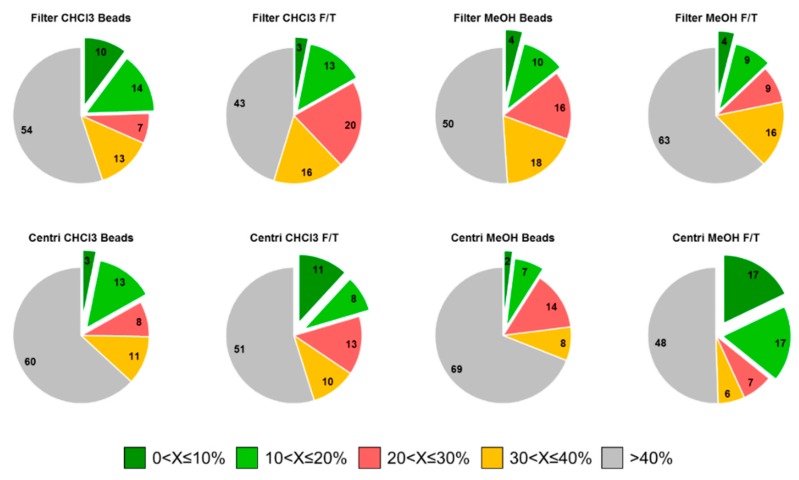
Relative standard deviation (RSD) values calculated on the area reached by each detected metabolite for the eight sample preparations investigated. RSD (X) are classified into 5 categories: 0 < X ≤ 10%, 10 < X ≤ 20%, 20 < X ≤ 30%, 30 < X ≤ 40%, or >40%.

**Table 1 metabolites-09-00193-t001:** Design of experiment (DOE) of the eight sample preparations, each investigated in triplicate, with differing cell retrieval systems (filter or centrifugation), quenching and extraction solvents (MeOH:H_2_O 8:2 or MeOH:H_2_O:CHCl_3_ 7:2:1 + EDTA 1 mM), and cell disruption mechanisms (bead beating or freeze-thaw cycles).

Filter/Centrifugation	Solvent MeOH/Solvent CHCl_3_	F/T Cycles/Beadbeating	Combination
+	+	+	Filter MeOH F/T
-	+	+	Centri MeOH F/T
+	-	+	Filter CHCl_3_ F/T
-	-	+	Centri CHCl_3_ F/T
+	+	-	Filter MeOH Beads
-	+	-	Centri MeOH Beads
+	-	-	Filter CHCl_3_ Beads
-	-	-	Centri CHCl_3_ Beads

**Table 2 metabolites-09-00193-t002:** Relative variability and block contributions of the AMOPLS model of the data acquired from the investigated biological samples. RSR: residual structure ratio, tp1-3: predictive components, to: orthogonal component.

Effect	Contribution	RSR	*p*-Value	tp1	tp2	tp3	to
Cell retrieval	27.6%	1.92	0.2%	96.7% ^1^	3.9%	1.8%	15.9%
Quenching/ extraction solvent	8.4%	1.17	0.1%	1.0%	81.8%	3.1%	26.7%
Cell disrupting mechanisms	7.0%	1.14	0.4%	1.0%	6.7%	91.6%	26.9%
Residuals	57.0%	1.00	N/A	1.2%	7.6%	3.5%	30.5%

^1^ The highest contribution for each component is reported in bold.

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
