# Peer review of "Choosing an Optimal Sample Preparation in Caulobacter crescentus for Untargeted Metabolomics Approaches"

_metabolites, 2019, doi:10.3390/metabo9100193_

Round 1

Reviewer 1 Report

The manuscript by Pezzatti et al. deals with the optimization of the sample preparation step for the unbiased metabolomic analysis of Caulobacter crescentus, a bacterial model. Sample preparation optimization is indeed a very important step in untargeted metabolomics, but it is often ignored. In this sense, the authors show an elegant way to conduct sample preparation optimization for the given matrix. The manuscript presented is well written and adjusts to the topic of the special issue. I personally found the ANOVA Multiblock OPLS algorithm employed here as a very interesting way of showing the effect on the variability coming from different sources evaluated during the sample preparation. I recommend publication of this manuscript after some minor issues are discussed/addressed.

In the experimental section it is mentioned that sample normalization was done, later on it is detailed that it is a PQN normalization. Did not the authors additionally normalize the metabolites abundancies to number of cells or protein content? This is a critical step in metabolomics of cells (Hounoum et al. 2016 TrAC 75, 118-128). Perhaps authors should discuss how this applies to the specific analysis on bacterial samples. Specially since there seems to be a lack of consensus in the metabolomics community about which one should be better to use for normalization, a critical comment from authors could be included in the manuscript.

Consistency on term quenching/extraction should be followed along the manuscript. For example, authors should consider to name section 2.2.2. as quenching/extraction methods and not just quenching.

Authors use negative ESI mode for the analysis. Can they provide more insight on why this mode and not positive ionization (widely used in metabolomics) was used?

Lines 204-207, authors are encouraged to provide references to backup the statement on why EDTA was added to CHCl3 containing extractant solvent and not to the first solvent.

Lines 226-227. It is not completely clear whether those 904 features were all commonly found in the eight experimental combinations. In this case, did authors evaluate what metabolites appeared new and which ones were in common when comparing the two different extraction solvents? Perhaps a Venn’s diagram with number of metabolites per solvent could be interesting to display the overlap and differences between them.

Lines 289-291. Can authors hypothesize what “other” sources might be responsible for more than 50 % unexplained variability included in the residuals of the AMOPLs model?

Layout on Figure S3 should be better explained. Perhaps authors should include what side (right or left) belong to what condition, for example in A panel what side belong to filter and which one for centrifugation.

Reviewer 2 Report

This research work deals with an issue relevant for metabolimics studies as it is the sample preparation, this including cell harvesting and metabolite extraction. The results are of interest for any working in the metabolimics field, and they are properly presented and discussed.

I recommend publication after a few concerns are addressed as follows:

1) Lines 117-122: why were 3 hours chosen for growing after cell collection by filtration? A comparison with fast extraction after cell collection would have been of interest as this procedure may have altered the cell metabolism due to changed nutrient access and cell grouping or distribution in the culture medium. As well, metabolite release might have been enhanced.

2) Why did the authors chose chloroform as extraction solvent? Chloroform is an acidic solvent and other more common solvents used in metabolomics studies would have been of higher interest as it is acetonitrile for example. Dichloromethane would have even been more appropriate to avoid the acidic effects.

3) Please provide the manufacturer of the chromatographic HILIC column used.

4) A final table or figure where the most appropriate method for each metabolite group is illustrated would be welcome.

5) A remark on the suitability of the sample preparation for any given analytical condition in the LC-MS platform should be taken into account as MS conditions may influence the results for different metabolite groups as well.
